# Managing the Risk of Foodborne Infections in Pediatric Patients with Cancer: Is the Neutropenic Diet Still an Option?

**DOI:** 10.3390/nu16070966

**Published:** 2024-03-27

**Authors:** Laura Pedretti, Davide Leardini, Edoardo Muratore, Gaia Capoferri, Serena Massa, Sofia Rahman, Susanna Esposito, Riccardo Masetti

**Affiliations:** 1Pediatric Clinic, University Hospital, Department of Medicine and Surgery, University of Parma, 43126 Parma, Italy; laurapedre@hotmail.it (L.P.); capoferrigaia@gmail.com (G.C.); serena.massa92@gmail.com (S.M.); sofia.rahman@outlook.it (S.R.); 2Pediatric Hematology and Oncology, IRCCS Azienda Ospedaliero-Universitaria di Bologna, 40138 Bologna, Italy; davide.leardini3@studio.unibo.it (D.L.); edoardo.muratore@gmail.com (E.M.); riccardo.masetti5@unibo.it (R.M.)

**Keywords:** cancer, foodborne infections, gut microbiota, hematopoietic cell transplantation, neutropenic diet

## Abstract

Infections pose a significant threat to morbidity and mortality during treatments for pediatric cancer patients. Efforts to minimize the risk of infection necessitate preventive measures encompassing both environmental and host-focused strategies. While a substantial number of infections in oncologic patients originate from microorganisms within their native microbiological environment, such as the oral cavity, intestines, and skin, the concrete risk of bloodstream infections linked to the consumption of contaminated food and beverages in the community cannot be overlooked. Ensuring food quality and hygiene is essential to mitigating the impact of foodborne illnesses on vulnerable patients. The neutropenic diet (ND) has been proposed to minimize the risk of sepsis during neutropenic periods. The ND aims to minimize bacterial entry into the gut and bacterial translocation. However, a standardized definition for ND and consensus guidelines for specific food exclusions are lacking. Most centers adopt ND during neutropenic phases, but challenges in achieving caloric intake are common. The ND has not demonstrated any associated benefits and does not ensure improved overall survival. Consequently, providing unified and standardized food safety instructions is imperative for pediatric patients undergoing hematopoietic cell transplantation (HCT). Despite the lack of evidence, ND is still widely administered to both pediatric and adult patients as a precautionary measure. This narrative review focuses on the impact of foodborne infections in pediatric cancer patients and the role of the ND in comparison to food safety practices in patients undergoing chemotherapy or HCT. Prioritizing education regarding proper food storage, preparation, and cooking techniques proves more advantageous than merely focusing on dietary limitations. The absence of standardized guidelines underscores the necessity for further research in this field.

## 1. Introduction

Infections pose a significant threat to morbidity and mortality during treatments for pediatric cancer patients [1]. Various factors, such as skin and mucosal damage, the use of central venous devices, prolonged or severe neutropenia, and concurrent immunosuppressant use, contribute to the increased risk of opportunistic infections in children undergoing cancer treatment [2]. Sepsis represents one of the leading causes of fatal complications, and neutropenia significantly elevates the risk of sepsis [3].

The mortality rate attributed to infection in pediatric patients with severe neutropenia is estimated to be approximately 0.4% to 1.0% in those receiving chemotherapy and experiencing febrile neutropenia [4]. An association has been suggested between febrile neutropenia and gut microbiota (GM) composition. Specifically, individuals with a more stable GM profile over time tended to experience a shorter duration of fever. Patients with prolonged fever had higher levels of certain GM microbes before hematopoietic cell transplantation (HCT). These results could suggest a potential link between GM composition and the development and progression of febrile neutropenia [5]. Preventive measures, including environmental and host-focused strategies, are needed to reduce the morbidity and mortality associated with infectious complications. While a significant proportion of infections in oncologic patients are believed to originate from organisms within the resident microbiological flora, the possibility of bloodstream infection (BSI) potentially acquired through the consumption of contaminated food and beverages in the community is a concern [2]. Foodborne infections are particularly challenging, causing gastroenteritis with symptoms such as diarrhea, vomiting, abdominal pain, and dehydration. Complications can be severe, affecting the nervous system and joints and causing hemorrhagic colitis, nutritional deficiencies, vascular issues, and renal failure [6,7,8,9]. Children are at a higher risk of enteric and foodborne diseases. Age-related factors (physiological impairment of the immune system and experimental touching and testing behaviors), dietary factors (tendency to consume certain types of foods such as dried milk products, snack foods, and candies), and environmental factors (direct contact with floors and other surfaces and the presence of an infected person or pet at home) are some of the main risk factors for the pediatric population [10]. Further risks extend to the young with pre-existing or underlying conditions that affect their natural immunity, such as children with oncohematological conditions [11,12,13,14,15,16]. Monitoring food quality and hygiene is vital to reducing the incidence and severity of foodborne illnesses [17].

Therefore, promoting food safety and educating patients on healthy eating practices is crucial to minimizing the risk of infections and their complications in vulnerable populations [18]. The neutropenic diet (ND), a low-bacteria diet, has been developed in recent years to reduce food-related risks [3]. The rationale behind the ND is to minimize the introduction of bacteria and potential pathogens from food into the gut, thus reducing the risk of bacterial translocation and bacteremia [19]. The ND generally involves excluding foods such as fresh fruits and vegetables and limiting dairy and meat products, especially the raw ones. However, there is no universal definition of the ND, and national consensus guidelines for specific food exclusions are lacking [20].

It is not only important to consider the food safety implications of different recommendations but also the impact on nutritional intake and sufficiency, for which there is a dearth of international guidelines [21]. Most centers have specific nutritional guidelines for transplant patients, and the majority adopt the ND during the neutropenic phase. The duration of dietary restriction is based on factors such as immunosuppressive therapy, time after transplant, and cellular immune status [22]. Patients undergoing HCT often face challenges in acquiring sufficient oral calorie intake due to the side effects of treatment, such as mucositis, nausea, vomiting, and diarrhea. Malnutrition can have long-lasting and serious effects on the transplant process and is considered an independent risk factor for morbidity and mortality in HCT patients [22,23]. The gut microbiota (GM) also plays a crucial role in the outcomes and side effects of HCT. Dietary nutrients significantly influence the GM’s diversity and function. Interestingly, some foods excluded from the ND may act as natural sources of probiotics and prebiotics, which promote a healthy and diverse intestinal microbial environment [20]. This narrative review aims to evaluate the impact of foodborne infections in pediatric cancer patients and the role of the ND in relation to food safety practices in patients undergoing chemotherapy or HCT. Electronic searches were conducted in the PubMed database from 1970 to 2023 using the following keywords: “neutropenic diet” OR “foodborne infection” AND “cancer” OR “hematological diseases” AND “children” OR “pediatric” OR “paediatric”. Only articles written in English were selected, and a manual reference search of the eligible articles was conducted.

## 2. Foodborne Infection in Children with Cancer or Undergoing Hematopoietic Cell Transplantation

There are various ways in which pathogens can contaminate food and drinking water. Contamination can occur at any phase of the farm-to-table process. For instance, it can occur due to pesticide use or soil contamination during the packaging process, food transportation, food handling, or the cooking process [17]. *Salmonella* spp., *Campylobacter* spp., Shiga toxin-producing *Escherichia coli*, hepatitis E virus, and norovirus are among the most common microorganisms involved. *Listeria monocytogenes* is responsible for some of the most severe infections, especially in immunocompromised patients [24,25]. When collecting the medical history of a sick patient, information regarding possible food exposure is a fundamental step, even though identifying the exact source of the disease is often difficult. Patients may find it difficult to recall the composition of their meals, especially when the exposure occurred days or weeks earlier. Additionally, it can be challenging to trace the origins of ingredients sourced from different places and used in a single dish. Moreover, not all people who consume contaminated food develop a symptomatic disease. Strict monitoring of animals and the food chain, the promotion of proper hygiene procedures such as thorough handwashing and proper administration of vaccines in animals, together with proper food cooking, preparation, and storage, are crucial to decreasing the incidence and severity of foodborne illnesses [17,26,27,28,29,30,31].

Foodborne infections are indeed rare in pediatric patients with oncological disease or undergoing HCT. Moreover, it is difficult to ascertain the incidence of foodborne diseases in this population due to the sporadic nature of cases reported in the literature and the authors’ frequent omission of the exact food source of the disease. Furthermore, there is no national registration system [26] that may enable the tracking of each infection. The most reported instances in the literature are summarized in Table 1.

In a retrospective study conducted in 2011, 21 bacterial infections were observed in 15 pediatric patients who underwent chemotherapy or immunosuppression due to a recent HCT. Out of these cases, only 3 (14%) were identified as foodborne. In one of these three cases, the origin of the infection was uncertain, while in the other two cases, the source of the food was not properly checked and investigated in patients with severe neutropenia [2]. Arora et al. report the case of a 15-year-old boy with relapsed ALL who developed a *Bacillus cereus* infection on induction chemotherapy. This pathogen predominantly contaminates cream-filled pastries, potatoes, eggs, or other salads with creamy dressings, but can also colonize catheters or open wounds. In this specific case, Arora et al. were not able to identify the specific source of the infection [16]. In addition, Saleeby et al. reported a case of bacteremia caused by *Bacillus cereus*. A 17-year-old girl with ALL in the induction phase of chemotherapy was infected. The authors of the study determined that the source of the infection originated from a non-sterile preparation of tea leaf extract. In fact, tests of tea bags taken from the patient’s room and from other locations in the hospital revealed a high prevalence of contamination by the Bacillus species [27].

In a large single-center study at the “Fred Hutchinson Cancer Research Center” in Seattle, Boyle et al. conducted a one-year follow-up study on all patients who underwent HCT between 2001 and 2011 [28]. The aim was to estimate the incidence of foodborne diseases in these patients. Patients were considered to have a bacterial foodborne infection if *Campylobacter jejuni*, *Listeria monocytogenes*, *Escherichia coli* 0157:H7, *Salmonella* spp., *Shigella* spp., *Vibrio* spp., or *Yersinia* spp. were isolated in culture within one year post-transplant. Again, the study failed to specify the infection’s actual food source. Of the 4069 patients, 12 (0.3%) developed a bacterial foodborne infection, and 3 of these were pediatric patients. The clinical presentation in the three children was severe, but all the patients recovered after adequate and prolonged antibiotic therapy and appropriate supportive care [28].

The viruses that are more commonly involved in foodborne diseases are norovirus and hepatitis E virus [29]. A prospectively enrolled surveillance study of diarrhea in 55 HCT patients and 61 solid organ transplant (SOT) recipients with a mean age of 9 ± 6.8 years showed that 25 (22%) were positive for the presence of norovirus in stool. Fifty percent of Norovirus patients experienced diarrhea lasting ≥14 days, and fifty-five were hospitalized for diarrhea, with 27% requiring admission to the intensive care unit (ICU). Therefore, noroviruses are the principal causes of diarrhea and are associated with significant complications in immunocompromised pediatric transplant recipients [30]. Furthermore, norovirus infection, which is often foodborne (fresh fruits and vegetables are typically the foods involved) or waterborne, causes chronic gastroenteritis due to the prolonged viral shedding [31].

Timely identification of the diarrhea’s etiologic agent is crucial for improving patient clinical management, such as by helping limit antibiotic use or avoiding invasive endoscopic procedures that are performed during the investigation of different diagnoses like graft versus host disease (GVHD) or adverse events resulting from immunosuppressive therapy [30]. Another virus involved in foodborne diseases is the hepatitis E virus. This foodborne infection, which can result from the consumption of undercooked game or pork meat, can lead to chronic hepatitis in immunocompromised organ transplant recipients [11,32]. In this subset of patients, chronic infections can be life-threatening, as hepatic insults can lead to inflammation and fibrosis. Furthermore, the need for re-transplantation as a result of post-transplant hepatitis is of great concern [33].

Finally, *Cryptosporidium*, *Giardia*, and *Toxoplasma* are common parasitic foodborne infections in children with cancer and other hematological conditions [11,15,34]. Domenech et al. report two cases of *Cryptosporidium* infection during maintenance chemotherapy for ALL [15]. It is well known that *Cryptosporidium* can cause prolonged watery diarrhea and malabsorption associated with nausea and vomiting that can be life-threatening in transplant patients; moreover, the organism is very difficult to eradicate [11,34]. Toxoplasmosis is a rare infection that can occur after HCT and SOT, but it has a significantly high mortality rate [11]. The reported cases are often associated with inconsistent use of the prophylactic trimethoprim and sulfamethoxazole (TMP-SMX) therapy, and the main sources of infection are undercooked meat, unpasteurized sheep or goat’s milk, contaminated vegetables, raw oysters, clams, mussels, and also contact with contaminated cat feces [35,36,37].

Given the predictability, potential severe clinical manifestations, and limited effective prophylactic therapies associated with these pathologies [38], it is understandable why significant efforts have been made to educate patients on the importance of consuming healthy and properly inspected food to reduce the risk of infections and their complications [18].

## 3. Dietary Restrictions to Reduce the Risk of Foodborne Infections: From a Neutropenic Diet to Safe Food Handling

Children receiving antiblastic therapy have an increased risk of developing neutropenia, defined as an absolute neutrophil count (ANC) less than 500 cells/mm^3^. Neutropenia significantly elevates the risk of potentially fatal infections, such as sepsis. A low-bacteria diet, referred to as ND, has been developed to reduce the risk of sepsis during periods of neutropenia [3]. ND, also known as the immune-compromised diet, low-bacteria diet, low-microbial diet, or sterile diet, involves excluding foods considered high in microbial risk, especially fresh fruits and vegetables, and limiting dairy and meat products. However, there is no universal definition for the ND, and there is a lack of consensus regarding specific food exclusions [20]. The rationale behind the ND is to reduce the introduction of bacteria into the gut and potential pathogens contained in food sources, thereby lowering the risk of bacterial translocation and bacteremia [19].

The diet’s evolution over the years is summarized in Figure 1.

The ND was first introduced in the 1960s through the “sterile diet,” a diet in which foods were sterilized in autoclaves or irradiated to eliminate bacterial loads. This diet was unpalatable to patients because it lacked taste; thus, it was eventually replaced by the “cooked food” diet, which is a non-sterile diet that sought to eliminate foods with a high bacterial load [39]. The current ND, or low-bacterial diet, was developed in the 1980s with the objective of limiting the presence of bacteria to less than 500 colony-forming units/gram of food. Only a small proportion, specifically 20% of meat and 30% of fresh fruit and vegetables, possessed this attribute and were suitable for consumption by neutropenic patients [40]. Individuals following a neutropenic diet tend to have lower intakes of protein, energy, fiber, and vitamin C compared to those on a regular diet. These restrictions in the neutropenic diet may worsen existing nutritional deficiencies, which is of particular concern since many children undergoing cancer treatment already experience micronutrient deficiencies [23]. Patients undergoing cancer treatment are already at a higher risk of malnutrition. Malnutrition reduces immune defenses and, consequently, increases the risk of infection. It can also lead to more severe treatment-related side effects and a poorer treatment outcome [23,41]. Malnutrition is also linked to impaired gut barrier function, which increases the risk of bacteremia. All these reasons call into question whether the risk of foodborne infection from consuming a normal diet outweighs the risks associated with reduced nutritional intake and malnutrition. Additionally, following a neutropenic diet may significantly impact the quality of life of patients [23]. Furthermore, a lack of uniformity has been found in the development and administration of the ND, as evidenced by a study conducted by Braun et al. that demonstrates the inconsistency in this diet’s application. In this study, the survey was sent to 1639 pediatric oncologists at 198 institutions that are members of the Children’s Oncology Group. Fifty-seven percent of oncologists used ND in their hospitals, but food restrictions and start and end requirements were very different from one center to another [42]. Studies have investigated whether the ND’s microbiological profile and bacterial load differed from those of other diets that can be prescribed in a hospital ward. Maia et al. analyzed 36 food samples, of which 18 were part of a normal hospital diet and 18 were part of the ND. Five samples were considered inadequate for ingestion because of a microbiological count above recommendations for B. cereus (n = 4) and coagulase-positive Staphylococci (n = 1). In conclusion, ND did not have a lower microbial load compared to the regular hospital diet. Furthermore, samples from the ND had less fiber and vitamin C, and no samples achieved the suggested daily intake of fiber, calcium, and iron [43]. This scarcity of fiber and vitamin C is due to the limited representation of fresh fruit and vegetables in the ND, as cooking processes can reduce nutrients and diminish nutritional quality [44]. Similar results were examined in a study conducted by Galati et al. Thirty-four food samples were analyzed: three varieties of vegetables and nine varieties of fruits in raw form and sanitized with chlorine solution (n = 12), raw and washed under running water (n = 12), and cooked (n = 10). All samples were negative for the presence of Salmonella spp., and no colonies of coagulase-positive Staphylococci were found. Vitamin C was studied in 20 samples. The findings revealed a mean loss of 38.9% of vitamin C in cooked food compared to raw food. These results confirm that there is no vast microbiological impact on raw food but rather a loss of nutrients in cooked food. A systematic review, published in the Cochrane Database, analyzed 619 studies investigating the effectiveness of a low-bacterial diet compared to a control diet in preventing infections in adult cancer patients undergoing chemotherapy-induced neutropenia. The authors concluded that there is no substantiating evidence to endorse the utilization of a low-bacterial diet to prevent infections and their associated consequences [4]. A few pediatric clinical trials were conducted over the years to study whether ND could effectively reduce the impact of infection in children undergoing chemotherapy. One of the first studies was conducted by Moody et al. in 2006 [45], in which the safety and feasibility of the ND were studied and compared to the food safety guidelines (FSGs) set by the Food and Drug Administration (FDA). FSGs are based on five major areas that recommend how food should be shopped, stored, prepared, cooked, and served to avoid excessive bacterial contamination. Some recommendations are general advice that should be included in any diet, such as not to consume expired foods or to wash hands before cooking; others are more specific, such as the refrigerator’s temperature and cooking temperature for meat. It is important to emphasize that these suggestions are extremely less stringent than the neutropenic diet. In Moody’s randomized study, there was no statistically significant difference observed in the two main outcomes, febrile neutropenia and ANC, between the two study groups. The mean nadir for each group was 615 × 10^9^/L and 286 × 10^9^/L for the food safety and neutropenic diet arms, respectively. If only those patients who became neutropenic were included, then the mean nadir for each group was 184 × 10^9^/L and 130 × 10^9^/L for the food safety and the ND groups, respectively. Before the study concluded, an ANC below 1000 × 10^9^/L was 13.7 days for the food safety diet group and 7.9 days for the ND group. The mean number of days with an ANC below 500 × 10^9^/L was 9.2 days for the food safety arm and 5.9 days for the ND arm. At the end of the study period, researchers conducted weekly interviews using the 24-h diet recall method with cancer patients and their parents, or alone if over the age of 16, to examine adherence to the diet and tolerability. Adherence to the FSGs was 99.99%, as opposed to 94.1% for those who had followed the ND. Nine out of ten patients reported no difficulties in following the FSGs. Some parents also found the guidelines to be helpful. The parents’ sole recommendations were to add more information about nutrients, calories, and convenience foods and to translate the guidelines into their native language. With regard to the ND, all nine patients in the study admitted to having difficulties with dietary restrictions. The difficulty in sustaining the diet for more than one cycle was primarily due to “too much to do” and “takeout foods”. Typically, patients failed to adhere to the ND because of the restriction on raw fruit and takeout food. The fact that the frequency of infections in the two groups was similar suggests that cancer patients should be allowed to increase their food quality [45]. In 2018, the same authors conducted a similar multicenter study that evaluated the effectiveness of ND restrictions on a larger population of pediatric cancer patients. A group of 150 patients was randomly assigned to the FDA-approved FSGs (n = 73) or ND + FSGs (n = 77). The primary outcome was to detect neutropenic infections; the second outcome was to study ND adherence and acceptability. Twenty-four (33%) of FSG patients and twenty-seven (35%) of ND + FSG patients developed neutropenic infections. The microbiological cause of the infections was not found in most hospitalized patients. Furthermore, there were no significant differences in symptoms at presentation, including gastrointestinal symptoms, in the average duration of neutropenia (24 days for the FSG group and 25 days for the ND + FSG group), the degree of neutropenia (71% for the FSG group and 74% of the ND + FSG group developed a grade 4 neutropenia), and the days in which the ANC was <500 × 10^9^/L (9.6 days for the FSG group and 10.5 days for the ND + FSG group). Diet adherence in the FSG group was higher in the ND + FSG group (99.26 ± 4.8% versus 92.64 ± 15.32%). This second group of patients admitted to making a significant effort to follow the ND and were less satisfied. These data confirm the previous study, showing that a liberalized diet significantly increases adherence compared to a neutropenic diet and requires less effort, with no significant increase in neutropenic infections [46]. Tramsen et al. investigated the real effectiveness of the restrictions on social contact, pets, and food (including ND) imposed on patients with AML receiving chemotherapy in line with the AML-BFM 2004 trial. A total of 339 patients were enrolled in this multicenter analysis. The main outcome was the analysis of infectious complications, wherein data were gathered from surveys conducted by the different centers participating in the study. Following the statistical analyses, researchers found no significant association between the previously mentioned restrictions and a decreased incidence of FUO, bacteremia, pneumonia, and gastroenteritis [47]. A recent consensus statement from the Italian Association of Pediatric Hematology and Oncology (AIEOP) and the Alliance Against Cancer’s Survivorship Care and Nutritional Support Working Group did not recommend the ND, as there is no evidence to suggest that it is better than regular diets with respect to safe food handling [48].

In conclusion, the ND has not been shown to be effective in reducing infections in children undergoing chemotherapy, although many centers still use it. Table 2 summarizes the major studies addressing ND in pediatric oncology patients. The ND is often associated with a lack of patient compliance and a lower quality of life. Most importantly, the real benefits of the ND over an unrestricted diet have not been convincingly demonstrated. Therefore, the application of the ND should be questioned [49].

## 4. Food Restrictions for Patients Undergoing Chemotherapy or Hematopoietic Cell Transplantation

Chemotherapy and HCT are characterized by a high risk of mortality and morbidity due to infectious and immune-related complications, and intensive supportive care is provided. The significance of maintaining optimal nutritional status in these pediatric patients is increasingly recognized, both in terms of their immediate post-transplant outcomes and their long-term development [20]. Nutritional support during chemotherapy or HCT encompasses the type of diet administered during conditioning chemotherapy, nutritional assistance during the neutropenic phase, and dietary considerations after re-alimentation and discharge [20]. Patients undergoing chemotherapy or HCT often struggle to achieve an adequate oral caloric intake due to treatment side effects, including mucositis, nausea, vomiting, and diarrhea. The ND’s significant dietary restrictions in terms of variety and types of foods eaten can exacerbate nutritional deficiencies, resulting in a reduction in the daily intake of protein, fiber, and vitamin C. The long-term outcome of a ND in children undergoing chemotherapy or HCT can vary based on various factors such as the type and severity of cancer, individual response to treatment, and adherence to dietary restrictions. Regular monitoring is essential to assess the effectiveness of the diet and address any nutritional concerns, including the impact on their quality of life.

Malnutrition may have persistent, serious effects on the course of treatment and transplantation [23,50]. A study showed that malnutrition before, during, and after HCT is an independent risk factor for morbidity and mortality in patients who undergo HCT [22]. There are no published studies comparing the quality of life of patients following an ND versus those following a general diet during HCT. Trifilio et al. reported that patients who experimented with both diets “emphatically stated that they preferred” the unrestricted diet because it was less restrictive in terms of food choices and allowed them to maintain an adequate caloric intake [51].

### 4.1. Food Restrictions during Hematopoietic Cell Transplantation

At present, there is a dearth of international guidance on protocols for the cornerstones of supportive care for patients undergoing HCT, including food restrictions. Due to the absence of precise guidelines on food restrictions, there is inconsistency in clinical practices among different HCT centers [21]. Many centers follow their own trust guidelines or a mixture of guidelines from various sources [23]. Peric et al. conducted a survey among the European Society for Blood and Marrow Transplantation (EBMT) centers about their current nutritional practices. The survey was conducted from June 2015 to January 2016. The majority of respondent EBMT centers (93%) still regard the ND as a standard of care. These centers use either the standard ND or the ND with food fortification and oral nutritional supplements [52].

In the survey conducted by Braun et al. investigating the administration of the ND to patients undergoing HCT, 84% of the respondents stated that they had started the ND upon hospital admission or at the start of the preparative regimen. In contrast, 14% of the respondents initiated the diet based on ANC levels (89% when ANC was <500/L, 5% when ANC was <1000/L, and 5% when ANC was <1500/L). Twelve percent of respondents stopped the diet upon discharge, 24% stopped the diet when the patients were no longer neutropenic, 35% stopped the diet day +100 post-transplant, and 29% stopped the diet at discontinuation of immunosuppression. The survey results not only showed variability in practices among different institutions but also within the same institution. This variability in responses highlights the lack of clear guidelines on the topic of ND in pediatric patients [42]. Morris et al. conducted a national cross-sectional survey in the UK on food safety guidance across specialist centers providing HCT or high-dose chemotherapy. They reported consistencies in relation to the ND, specifically concerning the avoidance of unpasteurized dairy products, raw or undercooked meat, and unpasteurized patè. Significant inconsistencies were observed in the responses concerning the source of water used in the medical wards and the consumption of unpeeled fruits and vegetables [23]. Toenges et al. used a questionnaire to investigate nutritional treatment after allogenic HCT (allo-HCT) in centers performing HCT in Germany, Austria, and Switzerland. All centers reported having specific nutritional guidelines for transplant patients, and most centers (86%) adopted the ND during the neutropenic phase. The duration of dietary restrictions was related to immunosuppressive therapy, time after transplant, and/or cellular immune status. In terms of food-associated infections, 18% of centers observed infections during hospitalization, while 64% reported at least one infection in the outpatient setting, demonstrating the association between unregulated foodborne exposures in outpatient environments and food-associated infections [22].

### 4.2. Effectiveness of the Neutropenic Diet in Children Undergoing Hematopoietic Cell Transplantation

There is currently no scientific evidence demonstrating the efficacy of the ND in preventing food-borne infections in adult or pediatric cancer patients undergoing chemotherapy or HCT. The available data on the HCT population primarily originate from studies conducted on adult patients, with limited information available for pediatric patients [21]. It is essential to reevaluate the use of the ND given its uncertain value and its potential negative influence on nutritional status and quality of life [53].

Trifilio et al. conducted a retrospective study to analyze infectious complications in a population of 726 adult patients undergoing HCT. Half of the patients followed an ND, which excluded fresh fruit and vegetables, raw and undercooked meat and cheese, cold smoked fish, raw grain products, and unpasteurized dairy products. The other half followed a general hospital diet, which did not include uncooked meat, fish, or unpasteurized dairy products. The study found significantly fewer microbiologically confirmed infections in the general diet group (*p* < 0.027), particularly after the resolution of neutropenia (*p* < 0.008). Urinary tract infections and gastrointestinal infections (vancomycin-resistant *Enterococcus faecium*, *Enterobacteriacea*, and *Clostridium difficile*) were more common in the ND group. Changes to the normal GM induced by the change in diet could have contributed to the emergence of pathogenic bacteria within the gastrointestinal tract and subsequent urinary tract infections. There was no difference between the groups in terms of mortality. The ND group had a longer length of stay after engraftment and a higher frequency of diarrhea. Acute grade II-IV gastrointestinal GvHD was more prevalent in the ND group at 60 days after transplantation (47% versus 20%; *p* < 0.11) [51]. A randomized, controlled prospective pilot study by Lassiter et al. evaluated the incidence of infection and nutritional status in 46 adult patients undergoing HCT, with 25 patients on an ND and 21 patients on a diet without restrictions from the first day of conditioning until engraftment. The study found no significant differences in infection rates or nutritional status between the two groups. The authors concluded that, since immunocompromised patients typically have a reduced oral intake, it is essential to avoid restrictions that further limit food choices. Many of the foods restricted in the ND are cool and odorless (for example, fresh fruits and vegetables), which are appealing to patients undergoing HCT and experiencing mucositis, nausea, or vomiting [54]. A random-effects meta-analysis by Sonbol et al. involved 1116 adult and pediatric patients, of whom 772 underwent HCT and were compared in terms of the ND and a regular diet. The analysis did not highlight any statistically significant difference between the ND and a regular diet in terms of major infection rates (relative risk [RR] 1.16; 95% confidence intervals [CI] 0.94 to 1.42) or bacteremia/fungemia (RR 0.96; 95% CI 0.60 to 1.53). ND was associated with a slightly higher risk of infection in patients undergoing HCT (RR 1.25; 95% CI 1.02 to 1.54). There was no difference in mortality between the ND and a regular diet (RR 1.08, 95% CI 0.78 to 1.50) [55]. In the study by Taggart et al., 102 pediatric patients undergoing HCT were divided into two groups: 49 patients followed the neutropenic diet, while 53 patients followed the modified HCT diet, which is a food safety-based diet with more relaxed restrictions. The study did not find substantial evidence indicating differences in the incidence of BSI, viral infections, GvHD, days of total parenteral nutrition use, or death in the first 100 days post-transplant. There were no significant differences in subjective tolerance between the two diets [50]. Despite the lack of scientific evidence in this regard, many centers still recommend the ND, viewing it as a cautious and sensible practice [19].

A recently published study conducted a prospective, multicenter, randomized phase 3 trial that included all consecutive hospitalized adult patients undergoing auto- or allo-HCT, as well as adult patients expecting a period of neutropenia >7 days. The study aimed to investigate the impact of a protective diet (involving foods cooked at >80 °C and washed and peeled thick-peel fruit) compared to a non-restrictive diet (involving washed fresh fruit and vegetables handled in accordance with safe food handling practices). A total of 222 patients were randomly assigned and analyzed, with 111 in the protective diet group and 111 in the non-restrictive diet group. The study found that the incidence of fever of unknown origin, including febrile neutropenia with unknown origin, and sepsis was similar in both arms. The incidence of acute GVHD was comparable between the two groups. Therefore, the restrictive diet poses an unnecessary burden on patients’ quality of life [56].

The role of diet has gained prominence as a significant factor in influencing the composition, metabolic profile, and resilience of the GM, particularly in the context of transplantation, due to the profound connection between the nutrients we consume and the makeup and functions of the GM. Alterations in diet, even in the short term, can induce rapid changes in the GM. A diet rich in fiber and fermented foods can enhance GM diversity, reduce inflammation markers, and exert a distinct influence on immune responses [57]. In patients undergoing allo-HCT, there is a destruction of the GM, characterized by reduced diversity and a prevalence of potentially pathogenic bacteria, mainly caused by the therapies such patients undergo for both the underlying pathology and the transplant procedure. GM dysbiosis in patients undergoing allo-HCT contributes to major adverse outcomes, such as the development of acute GVHD and the occurrence of BSI, and correlates with mortality [20]. Furthermore, nutritional support is increasingly recognized as a crucial element in modulating the GM of allo-HCT recipients. Nutrients introduced through the diet are major modulators of the GM’s diversity and function. Foods commonly excluded from the ND may be natural sources of probiotics and prebiotics that promote intestinal microbial diversity [20,23]. Increasing attention to the impact of nutritional status and gut dysbiosis on the clinical outcome of cancer patients and improvements in supportive care have enabled a reevaluation of the use of the ND [19].

### 4.3. Food Management Practices

Recently, the EBMT, the European Society for Clinical Nutrition and Metabolism (ESPEN), and the American Health Organization proposed replacing the ND with the implementation of safe food handling guidelines. This would result in enhanced palatability of foods and, consequently, an increase in caloric intake without raising the risk of infection [51,58,59,60]. The U.S. Department of Agriculture/FDA still advises against the consumption of raw ground meat or unpasteurized milk and dairy products, classifying other foods such as fresh vegetables and salad as low-risk foods for cancer patients as long as certain handling and preparation conditions are strictly adhered to [21]. Fresh, raw fruits and vegetables are permitted if previously washed under running tap water to ensure they are free from visible damage [20].

The intake of fiber-rich fruits and vegetables may be beneficial for GM homeostasis by providing the substrate for SCFA production, consequently modulating mucosal immunity, and reducing the incidence of GVHD [61,62]. Ifversen et al. proposed an algorithm to provide guidelines on protective measures for pediatric and adolescent HCT patients after they are discharged to home. These guidelines are based on expert consensus and include food restrictions, social distancing, and behavior at home and away from home.

Patients are stratified into three risk classes—low, intermediate, and high—based on immune recovery, the presence of GVHD, the level of immunosuppressive therapy, and infectious frequency. This assessment is performed at discharge, at day +100, and at day +180 post-HCT [21]. Clean, separate, cook, and chill are the four essential steps to be followed to reduce pathogen overgrowth in food: “clean” brings together detailed recommendations for washing hands and surfaces; “separate” provides guidance on how to prevent cross-contamination from one food to another; “cook” provides guidance on how to cook different foods at safe temperatures, and “chill” provides guidance on how to properly refrigerate foodstuffs [21].

## 5. Conclusions

Pediatric cancer patients face an elevated risk of infectious complications, with a notable proportion of infections arising from organisms within the resident microbiological flora. Although infrequent, foodborne infections can lead to severe clinical manifestations [2,38]. Given their susceptibility to such illnesses, it is imperative to educate cancer patients in order to minimize risks and improve treatment outcomes [25]. Various dietary approaches, including the ND, have been developed to mitigate the risk of infections. The ND has failed to demonstrate any associated benefits, nor does it ensure superior overall survival [63]. Despite challenges, ND has been widely administered to both pediatric and adult patients as a precautionary measure [4,59]. Therefore, it is essential to offer shared and standardized food safety guidance to pediatric patients undergoing chemotherapy and HCT [23,46]. Clinical trials are ongoing, and long-term outcome analysis as well as data on the effects on pediatric cancer patients’ health, physical development, and quality of life are urgently needed. Emphasizing education about proper food storage, preparation, and safe cooking practices rather than focusing solely on food restrictions positively impacts both the quality of life and the GM [23]. Given the documented influence of various dietary elements on GM, it would be interesting to investigate the impact of specific foods on intestinal flora and HCT outcomes [20]. It would also be useful to evaluate the necessity of differentiated dietary guidelines based on the type of transplant the patient undergoes, taking into account the potential significant differences in the intensity of chemotherapy and immunosuppressive treatment [23]. The absence of standardized guidelines underscores the necessity for further research in this domain [64].

## Figures and Tables

**Figure 1 nutrients-16-00966-f001:**
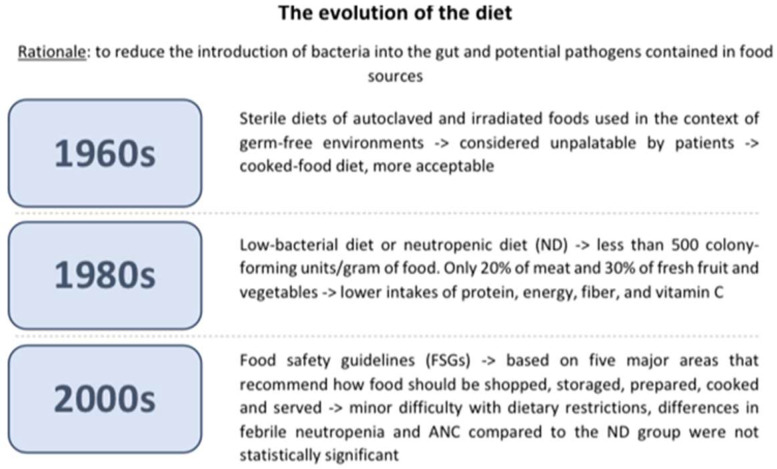
The diet’s evolution over the years.

**Table 1 nutrients-16-00966-t001:** Main studies on foodborne infection in children with cancer or undergoing hematopoietic cell transplantation (HCT).

Title of the Study, First Author, Year [ref.]	Type of Study	Objective	Population	Cases	Stage of the Disease at Time of Infection	Type of Chemotherapy	Type of Infection, Source	Clinical Symptoms	Outcome
Food-borne bacteremic illnesses in febrile neutropenic children. Anselm Chi-Wai Lee, 2011 [2]	Three-year retrospective hospital chart survey	To review all cases of documented bacteremia associated with febrile neutropenia * at the Children’s Hematology and Cancer Center of Singapore from March 2007 to February 2010	15 children with a solid tumor or leukemia	(A) A 17-year-old boy with osteosarcoma (B) A 2-year-old boy with ALL (C) A 2-year-old girl with ALL	(A) Induction (B) Induction (C) New diagnosis	(A) Ifosfamide and etoposide (B) 6-mercaptopurine (after treatment with cyclophosphamide)	(A) *Sphingomonas paucimobilis* bacteremia from a nasi lemak meal (B) *Chryseobacterium meningosepticum* bacteremia from a sushi dinner (C) *Lactobacillus spp.* bacteriemia suspected to be of probiotic origin	(A) Fever, vomiting, chills, rigors, and septic shock (B) Cyanosis and septic shock (C) Fever	Recovery
Cryptosporidiosis in children with acute lymphoblastic leukemia on maintenance chemotherapy. Carine Domenech, 2011, [15]	Case report	To describe two cases of children on maintenance chemotherapy with severe diarrhea disease caused by *Cryptosporidium*	2 children with ALL	(A) A 13-year-old boy with a late combined ALL relapse (8 years after the first complete remission) (B) A 2½-year-old boy with T-cell ALL with hyperleukocytosis, complicated by a massive intracranial hemorrhage before treatment onset. **	(A) Maintenance (B) Maintenance	(A) Thioguanine and methotrexate (B) 6-mercaptopurine and methotrexate	*Cryptosporydium* gastroenteritis, definite source of the infection is NA	(A) Severe diarrhea, severe weight loss, and weakness (B) Profuse watery stools, severe anorexia, and repeated vomiting with blood. The patient later developed cholangitis	Recovery
Bacillus cereus infection in pediatric oncology patients: A case report and review of literature. Sunisha Arora, 2021, [16]	Case report and systematic review	To describe the case of a child on induction chemotherapy who developed *Bacillus Cereus*’s septicemia.	1 child with ALL	A 15-year-old boy	Induction	NA	*Bacillus Cereus* septicemia, definite source of the infection is NA	Hematemesis, difficulty breathing (Oxygen saturation 92% on room air), drowsiness, and severe hypotension (BP 70/30 mmHg). The patient later developed encephalopathy, DIC, and multiorgan dysfunction.	Death
Association between tea ingestion and invasive Bacillus cereus infection among children with cancer. El Saleeby CM, 2004, [27]	Case-control study, prompted by a clinical case	To demonstrate an association between dietary tea ingestion and *B. cereus* bacteremia	1 child with ALL	A 17-year-old girl	Induction	NA	*B. cereus* bacteremia from a tea bag made from *Camellia Sinensis* leaves	NA	Recovery
Quantitative detection of norovirus excretion in pediatric patients with cancer and prolonged gastroenteritis and shedding of norovirus. Ludwig A, 2008, [31]	Retrospective cohort study	To describe noroviral shedding in immunocompromised patients and its correlation with clinical symptoms	9 children with a solid tumor or leukemia	(A) A 1½-year-old boy with rhabdomyosarkoma (B) A 2½-year-old girl with ALL and trisomy 21 (C) A 15½-year-old girl with JMML with PTPN11 mutation (D) A 1½-year-old boy with ALL and trisomy 21 (E) A 7-month-old boy with pre-B ALL (F) A 6½-year-old boy with neuroblastoma (G) A 2-year-old boy with c-ALL (H) An 8-month-old boy with neuroblastoma (I) A 5-month-old boy with AML M5	(A) 2a, N0, M0 (B) NA (C) NA (D) NA (E) NA (F) Stage IV (G) NA (H) Stage IV (I) NA	(A) CWS-2002 high-risk protocol (B) COALL 07-03 low-risk standard protocol (C) AML-BFM 2004 protocol (D) COALL 07-03 high-risk standard protocol (E) Interfant-99 standard risk protocol (F) NBL-97 high-risk protocol MIBG (G) COALL 07-03 high-risk standard protocol (H) NBL-97 high-risk protocol (I) AML-BFM 2004 high-risk protocol	*Norovirus* gastroenteritis, definite source of the infection is NA	Vomiting (66.6%), fever (44%), and diarrhea (44.4%)	Recovery
Noroviruses as a cause of diarrhea in immunocompromised pediatric hematopoietic stem cell and solid-organ transplant recipients. Ye X, 2015, [30]	Prospectively enrolled epidemiologic surveillance study	To evaluate the prevalence and clinical significance of *Norovirus* diarrhea among pediatric transplant recipients at *Texas Children’s Hospital* in Houston, Texas	116 children (61 SOT and 55 HCT recipients)	/	Post SOT or post HCT	NA	*Norovirus* gastroenteritis, definite source of the infection is NA	Severe diarrhea (lasting on average > 14 days and requiring frequent hospitalization), severe dehydration and hypovolemia (n = 2), respiratory distress (n = 2), septic shock (n = 1), cardiac arrhythmia (n = 1), and pneumatosis intestinalis (n = 1)	Death for three patients and recovery for the others
Bacterial foodborne infections after hematopoietic cell transplantation. Boyle NM, 2014, [28]	Retrospective cohort study	To describe the incidence of bacterial foodborne diseases after HCT	4069 HCT recipients (children and adults)	(A) A 4-year-old girl with CIMMDIS (B) A 17-year-old girl with CML (C) A 3-year-old boy with neuroblastoma	Post HCT	(A) Cyclosporine and mycophenolate mofetil (B) Clophosphamide and busulfan (C) Melphalan, etoposide, and carboplatin	(A) *Campylobacter* gastroenteritis (B) *Salmonella* gastroenteritis (C) *Salmonella* gastroenteritis and bacteremia Definite sources of the infections are NA	NA Patients A and B developed gut graft-versus-host disease	Recovery

* All patients were neutropenic at the time of the infections; ** as the patient responded poorly to the corticosteroid pre-phase, he was treated in the very high-risk group of the protocol European Organization for Research and Treatment of Cancer 58,881. NA = not available; HCT = hematopoietic cell transplantation; ALL = acute lymphoblastic leukemia; JMML = juvenile myelomonocytic leukemia; DIC = disseminated intravascular coagulation; AML = acute myeloid leukemia; CIMMDIS = immune deficiency disorder; CML = chronic myeloid leukemia; SOT = solid organ transplantation.

**Table 2 nutrients-16-00966-t002:** Major studies addressing neutropenic diet (ND) in pediatric oncology patients.

Title of the Study, First Author, Year [Ref]	Type of Study	Objective	Population	Outcome
Feasibility and Safety of a Pilot Randomized Trial of Infection Rate: Neutropenic Diet Versus Standard Food Safety Guidelines, Moody K, 2006 [45]	Multicenter prospective randomized controlled trial	To evaluate the infection rate in pediatric cancer patients randomized to the ND or FDA-approved FSGs and assess tolerability and adherence to the diets.	19 pediatric oncology patients (ND n = 9; FSGs n = 10)	Infection rates for children on the ND were similar to those of patients following FSGs; the adherence rate was 94% for the neutropenic diet and 100% for the food safety guidelines.
A randomized trial of the effectiveness of the neutropenic diet versus food safety guidelines on infection rate in pediatric oncology patients, Moody K, 2018 [46]	Prospective randomized controlled trial	To study neutropenic infection rates in pediatric oncology patients randomized to FSGs versus the ND plus FSGs; study adherence to the diets and acceptability.	150 patients were randomly assigned to FSGs (n = 73) or ND + FSGs (n = 77)	ND offers no benefit over FSGs in the prevention of infection; diet adherence in the FSGs group was higher than in the ND + FSGs group.
Lack of effectiveness of neutropenic diet and social restrictions as anti-infective measures in children with acute myeloid leukemia, Tramsen L, 2016 [47]	Multicenter analyses on AML-BFM 2004	To check the effectiveness of non-pharmacological measures, including ND.	339 patients treated in 37 institutions.	Dietary restrictions were not significantly associated with a decreased incidence of FUO, bacteremia, pneumonia, and gastroenteritis.

ND = neutropenic diet; FDA = Food and Drug Administration; FSGs = food safety guidelines; AML = acute myeloid leukemia; BFM = Berlin-Frankfurt-Münster; FUO = fever of unknown origin.

## Data Availability

Not applicable.

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
