# Peer review of "Managing the Risk of Foodborne Infections in Pediatric Patients with Cancer: Is the Neutropenic Diet Still an Option?"

_nutrients, 2024, doi:10.3390/nu16070966_

Round 1
Reviewer 1 Report
Comments and Suggestions for Authors
The manuscript by Laura Pedretti and colleagues thoroughly examines the neutropenic diet (ND) for pediatric cancer patients at risk of infection from foodborne sources due to their compromised immune systems. The commentary questions the continued effectiveness of the ND as a preventive measure, as the literature review cited suggests a lack of consensus on its superiority over regular food safety guidelines.
The article also delves into the potential impact of the ND on the gut microbiota for pediatric patients undergoing hematopoietic cell transplantation (HCT). It raises concerns about potential nutritional deficiencies arising from the limitations of the ND, particularly in terms of patient outcomes and quality of life.
However, the paper may overlook a few points:
- Lack of Long-Term Outcome Analysis: It focuses on short-term outcomes without addressing potential long-term implications of dietary choices on pediatric cancer patients' health and physical development.
- Nutritional Deficiency Analysis: While it mentions nutritional deficiencies due to the ND, it lacks a detailed analysis of their long-term health implications for pediatric patients.
- Quality of Life Considerations: There may be a lack of in-depth analysis on how dietary restrictions emotionally and socially impact pediatric patients and their families.
In conclusion, the manuscript argues for a reassessment of dietary guidelines for pediatric oncology patients, emphasizing the importance of evidence-based practices and the potential need for updated strategies that may better serve patient health and recovery.
Author Response
The manuscript by Laura Pedretti and colleagues thoroughly examines the neutropenic diet (ND) for pediatric cancer patients at risk of infection from foodborne sources due to their compromised immune systems. The commentary questions the continued effectiveness of the ND as a preventive measure, as the literature review cited suggests a lack of consensus on its superiority over regular food safety guidelines.
The article also delves into the potential impact of the ND on the gut microbiota for pediatric patients undergoing hematopoietic cell transplantation (HCT). It raises concerns about potential nutritional deficiencies arising from the limitations of the ND, particularly in terms of patient outcomes and quality of life.
Re: Thank you very much for your comments. We revised the manuscript accordingly.
However, the paper may overlook a few points:
- Lack of Long-Term Outcome Analysis: It focuses on short-term outcomes without addressing potential long-term implications of dietary choices on pediatric cancer patients' health and physical development.
Re: Comments on the need of long-term outcome analysis have been added (pp. 12 and 16).
- Nutritional Deficiency Analysis: While it mentions nutritional deficiencies due to the ND, it lacks a detailed analysis of their long-term health implications for pediatric patients.
Re: This issue has been considered (p. 12).
- Quality of Life Considerations: There may be a lack of in-depth analysis on how dietary restrictions emotionally and socially impact pediatric patients and their families.
Re: Added (pp. 12 and 16).
In conclusion, the manuscript argues for a reassessment of dietary guidelines for pediatric oncology patients, emphasizing the importance of evidence-based practices and the potential need for updated strategies that may better serve patient health and recovery.
Re: Thank you for your comments and we hope that you could accept the manuscript in its revised version.
Reviewer 2 Report
Comments and Suggestions for Authors
Please change the ref recent ine.
Please delete the number 64 in ref.
Table 1 is not readable. Please change to readable.
Please include other cases besides hct.
Please include various clinical trials.
Author Response
Please change the ref recent ine.
Re: Done.
Please delete the number 64 in ref.
Re: Done (p. 19).
Table 1 is not readable. Please change to readable.
Re: We think that the Table is readable. In any case, we added it in word so that the Editor could try to add it in the text better than us.
Please include other cases besides hct.
Re: We clarified that we considered not only children undergoing HCT but also those treated with chemotherapy (pp. 1-4). Tables 1 and 2 summarize studies in all the oncologic pediatric patients. However, some paragraphs focus only on HCT because in this condition there are some specific implications.
Please include various clinical trials.
Re: Added, including further priorities for research (p. 12).